# Development and Testing of the Smart Healthcare Prototype System through COVID-19 Patient Innovation

**DOI:** 10.3390/healthcare11060847

**Published:** 2023-03-13

**Authors:** Po-Chih Chiu, Kuo-Wei Su, Chao-Hung Wang, Cong-Wen Ruan, Zong-Peng Shiao, Chien-Han Tsao, Hsin-Hsin Huang

**Affiliations:** 1College of Management, National Kaohsiung University of Science and Technology, Kaohsiung 824005, Taiwan; 2Department of Information Management, National Kaohsiung University of Science and Technology, Kaohsiung 824005, Taiwan; 3Department of Otolaryngology, Chung Shan Medical University Hospital and School of Medicine, Taichung 40201, Taiwan

**Keywords:** COVID-19, smart healthcare system, usability, prototype, patient innovation

## Abstract

Since the outbreak of the novel coronavirus disease 2019 (COVID-19), the epidemic has gradually slowed down in various countries and people’s lives have gradually returned to normal. To monitor the spread of the epidemic, studies discussing the design of related healthcare information systems have been increasing recently. However, these studies might not consider the aspect of user-centric design when developing healthcare information systems. This study examined these innovative technology applications and rapidly built prototype systems for smart healthcare through a systematic literature review and a study of patient innovation. The design guidelines for the Smart Healthcare System (SHS) were then compiled through an expert review process. This will provide a reference for future research and similar healthcare information system development.

## 1. Introduction

The outbreak of the novel coronavirus disease 2019 (COVID-19) changed the entire world. As of January 2023, more than 750 million people have been diagnosed, nearly 7 million have died from the outbreak, and the world economy has been devastated [1]. Even though many countries have lifted restrictions, it is expected that the global economy, and the lives of individuals, will continue to be affected. Given the speed and impact of epidemic diseases such as COVID-19, smart healthcare systems can effectively assist healthcare workers, reduce healthcare resource consumption, and assist healthcare teams and government agencies in building healthcare ecosystems to assist in policy decision-making. It is important to discover and understand the needs of system users through participatory research, and then evaluate and adjust the design of the system to optimize its usability [2,3]. During this epidemic, people have seen their lives change dramatically overnight, but more and more people are willing to devote themselves to their professions as amateur data scientists, using relevant technologies and data graphical analysis to assist in the management of medical decision-making.

In the past, information system design and development were mainly applied to various aspects of product and service process design. Ulrich focused on system design and development in the design of system functions [4] and aesthetic style [5]. Concurrently, companies were thinking about extending the design and development of information systems by introducing a human-centered approach to these R&D design processes. This approach needs to focus on the needs of system users and their use process rather than the attributes of the system itself, which is called ‘design thinking’ [6]. In recent years, there has been an extensive introduction into the healthcare field of an emphasis on patient-centeredness and patient engagement, as in the studies by Roberts et al., Oliveira et al. and Cennamo et al. [7,8,9]. Cennamo et al. suggested that innovation in the healthcare system can be facilitated by commercializing desired products via the empowering of care recipients through a multilateral platform innovation organization [9]. The following case studies are provided as examples to illustrate the process of how healthcare services were improved.

In the case of the Patient Innovation Platform, patient Catherine Patton was diagnosed with diabetes during her pregnancy in 2001, and her doctor required her to take insulin injections multiple times a day, like a diabetic, but this was a painful routine and she was left searching for possible solutions. To improve the experience of this treatment process, she proposed the development of a device that would solve the problem of repeating her painful experience every day. Such a product is designed to allow patients to inject insulin and other medications without having to pierce the skin to complete the injection. The product is designed to be worn for three days and can be used to meet the patient’s normal lifestyle needs, including sleeping, bathing, and physical training, while wearing the product [10].

According to a study by the Cliver Research Team at Carnegie Mellon University’s School of Design, the team was commissioned by the Department of Neurosurgery at the University of Pittsburgh Medical Center in 2007 to improve patient experience. The design team began by identifying all the steps in the patient’s journey before and after surgery to create a detailed customer service blueprint that illustrated the steps the patient would go through. The design team then shared their thoughts on the medical process through on-site observations and invited patients and the medical team to share their thoughts in order to help the design team to understand the patient’s true feelings. Based on the provided information, the design team then analyzed the services needed by the patient and improved the patient experience [11].

COVID-19 has severely impacted and changed human habits worldwide since 2019. Soon after, different resources were invested in addressing the problems associated with COVID-19. While the disease may be under control, a review of recent human history shows that the frequency of such major diseases has been shortened by the ease of transportation. We should indeed prepare for a rainy day and use the lessons of COVID-19 to consider how to improve today’s smart healthcare system. Based on the spirit of human-centeredness, this study examines how innovative information technologies such as artificial intelligence can be incorporated into smart healthcare systems, and considers how to incorporate the concept of patient innovation, good communication between doctors and patients, and overall program evaluation. Therefore, the following research questions and definitions were developed.

How to promote patients’ participation in the integration of innovative technology and patient innovation.How to assist patients to use specific healthcare functions by combining their medical knowledge.How to help healthcare organizations effectively evaluate the usability and feasibility of these healthcare improvement projects.

We will examine the target users of the smart healthcare system, including the general public and healthcare workers. The general public can be subdivided into three user groups according to their needs: the first group consists mainly of those who use the system for their healthcare, the second group consists of those who use the system for their family members, and the third group of those who want to obtain relevant information through the system for follow-up analysis. As mentioned above, different user groups with different needs may have different evaluations of a smart healthcare system according to these needs. Accordingly, the following four research objectives are summarized.

4.To compile literature on the application and evaluation of artificial intelligence-based innovative technology applications in Smart Healthcare Systems (SHSs).5.To understand and analyze the actual usage requirements through patients’ participation and to assist patients in the design of the prototype system.6.To evaluate the use of the prototype system through usability testing and compile suggestions for system improvement from the participants in the process.7.To iterate the above process, reflecting on the needs and suggestions of different stakeholders while assisting patients’ participation.

COVID-19 virus is highly infectious and has high variability [12,13]. This study presents the outcomes of an industry-academia cooperation project. The university (National Kaohsiung University of Science and Technology) came up with the idea of developing an innovative SHS based on previous studies and on the current situation. Experts from the Chung Shan Medical University Hospital provided their suggestions from their professional point of view. This study aims to combine smartphones and webcams with artificial intelligence to enable healthcare practitioners to provide remote care to people in home quarantine. The researchers hope that the study can reduce the risk of contact for healthcare practitioners and improve their work quality and efficiency.

## 2. Materials and Methods

This study uses the grounded theory as the developmental framework of the research model and applies it to COVID-19 isolated patients in the SHS, as the contours of the research context. The grounded theory is a theory-generating method that follows the collection and analysis of data in the research process [14]. It is mainly applied to reveal social relationships and group behavior [15]. In Beech et al.’s study, the recovery experience of patients was observed through a surgical intervention experiment [16]. The three phases of this research process were constructed by using past research to assist in patient innovation, prototype system establishment, and usability testing and evaluation (Figure 1). In the first phase, we presented the background of the research on the use of artificial intelligence and other innovative technologies in an SHS, and examined possibilities for application in the usage context through the compilation of relevant literature. In the second phase, we invited COVID-19 patients or those with previous experience in caring for COVID-19 patients to conduct the interviews. In addition to listening to their experiences during treatments or quarantine, we also exchanged ideas on the use of innovative technologies in healthcare systems as inputs to the design of the prototype system and then completed the prototype design. In the third phase, we invited some of the respondents who had participated in the previous phase and nine experts from different fields of expertise to conduct a usability test of the system from the context of use requirements, and we compiled questions and corresponding suggestions from the experts about their thoughts on the system’s use flow and interface design. These suggestions and recommendations not only help the research team to improve the design of the prototype system but also serve as guidelines for the design of future SHSs. The final results and the knowledge and experience gained during the development process will be finalized. The results will be used as a reference for future research.

### 2.1. Phase I: Systematic Review of Recent Related Studies

We conducted a systematic review to identify the main trends in the development of user interfaces (UI) and user experiences (UX) in the SHSs. The literature was collected through ScienceDirect Online (SDOL), IEEE Xplore and MDPI journal databases. To have a clearer understanding of how the designers of the SHSs in the literature designed the UI/UX, the literature used in the study needed to be available in full text. In considering the rigor of the relevant research, the used literature must have been peer-reviewed, excluding conference papers with questionable research frameworks. Research articles related to SHSs were accepted during the systematic review process. We excluded papers on healthcare policy, vaccines, and treatment planning. The systematic literature review process follows the Preferred Reporting Items for Systematic Reviews and Meta-Analysis (PRISMA) guidelines [17]. Finally, the researcher has extended the focus from these primary references to related projects and studies for prototype system design. We compiled this information as input for the next phase of the study.

### 2.2. Phase II: Human-Centered Smart Healthcare System Development

Human-centered design is an approach to system design and development that focuses on the operational needs of the system to make it more usable and useful. Such an approach enhances efficiency and effectiveness, improves user satisfaction and sustainability, and reduces the potential adverse effects of system operation [18,19]. The concept of patient innovation even invites patients or their caregivers to participate directly in the development of the system. Patient innovation is a definition of the need for use by patients and/or the family and friends who take care of patients [3,9]. Therefore, this study recruited patients with related disease problems, through medical institutions or universities, to participate in research development. The prototype concept of the SHS is proposed by combining the literature collected in the first phase with patients’ personal experiences. An operational prototype system based on these concepts was constructed by the researchers for usability testing.

### 2.3. Phase III: Usability Test of the Smart Healthcare Prototype System

The prototype smart healthcare system developed is based on the Best of Both Worlds (BoB) framework as the design criteria for iterative development [20]. Thirty participants took part in the previous phase of interviews. Nine experts with more than 10 years of experience in their respective fields were invited to conduct the usability test for this study. These experts not only have rich working experience but also their independent views and in-depth insights into this study. The expert panel consists of three areas: three medical healthcare professionals, three information systems professionals, and three human factors professionals. The selection of the study participants was conducted following the guidelines of the Helsinki Declaration and was approved by the National Cheng Kung University Governance Framework for Human Research Ethics (NCUK HREC-E-109-571-2).

## 3. Results

In this section, we describe the information obtained through the selected papers. Through the systematic review, the researchers invited COVID-19 patients or those with experience caring for COVID-19 patients to conduct in-depth interviews. In addition to listening to these treatment and isolation care experiences, the researchers shared the information obtained in the first phase and exchanged ideas with the participants as inputs for the design of the prototype system. The researcher then compiled these design inputs and selected items of interest and suitability to complete the prototype system design. Finally, usability tests were conducted to understand the views of system users and experts and to suggest system improvements. The researchers iteratively repeated the above three phases, with each complete cycle running for one year and the complete study running for two years. The researchers then compiled the results of these usability tests and studies as a reference for future research.

### 3.1. Results of the Systematic Review

The systematic review uses a reproducible research methodology to compile existing published literature and to identify research findings relevant to the topic. In the process of compiling the relevant literature, the eligibility criteria we used to select the literature were as follows:Only available in English.Published between January 2019 and October 2022.Papers that discuss SHSs.

We excluded articles or papers that met the following criteria:Short conference papers.Full-text not available.Not related to healthcare information systems.

The following databases were used: ScienceDirect Online (SDOL), IEEE Xplore, and MDPI, to identify and collect articles related to SHSs. The selection was performed based on relevance to the domains of interest and scope. The fields considered in the search query were limited to the titles and abstracts of the papers. Several keywords (COVID-19, healthcare, systems, design) were used, and combined using Boolean operators (AND, OR, and NOT) to cross-examine the scientific databases. After retrieving the articles from the search databases, we use Endnote software to create references and remove duplicates. As highlighted in the inclusion criteria, articles were selected based on a three-step process: assessment of the title, abstract, and full text. The full process (as shown in Figure 2) is used for selection, including screening and determining eligibility and inclusion. The selected 22 documents in this phase can be divided into the following three categories, as shown in Table 1’s summary of the applications and the sub-topics covered.

#### 3.1.1. Internet of Medical Things (IoMT)

The Internet of Medical Things (IoMT) is the set of medical products and services that can be connected to medical and healthcare systems through other systems and networks [43]. Specific IoMT applications include telemedicine through wearable devices for patients with chronic diseases [32], and the ability of healthcare professionals to track patient prescriptions and providers through IoMT technology [44]. With a new IoMT integrated healthcare system, better personalized healthcare can be achieved. The benefits of the Internet of Medical Things also come with risks, such as serious information security issues and issues related to patient privacy [45].

The IoMT framework addresses the shortage of resources faced by medical teams during the COVID-19 epidemic; not only can the geo-location of COVID-19 patients be tracked by Bluetooth to assess personnel risk [26,27,31,40], but the technology allows for tracking of patient recovery [22,33]. Chen’s research focuses on visualizing and discussing this important information [35]. Through the presentation of these visual digital dashboards, public health units and government departments can quickly plan and allocate healthcare resources [36,39]. Gerli’s research focuses on analyzing the 50 e-health applications proposed by the European Union during the COVID-19 epidemic and outlines three issues (i.e., user orientation, participation, legality and equity) [23]. Patients and healthcare professionals can also learn online via virtual reality through IoMT technology [29,38]. During the COVID-19 epidemic, information security concerns about data breaches plagued the healthcare industry. Blockchain technology not only accelerates the adoption of electronic health records but also ensures that systems are resistant to malicious cyber-attacks [21,24,28].

#### 3.1.2. Clinical Decision Support System (CDSS)

CDSS can be defined as follows: “Health observation and health knowledge system are linked together to impact health choice by medical experts to enhanced healthcare system” [46]. The functions and benefits of CDSS include patient safety, clinical management, cost containment, administrative functions, and diagnostics support. Support for CDSS continues to grow in the age of the electronic medical record, and there are still more advances to be made including interoperability, speed and ease of deployment, and affordability [47]. CDSSs can help healthcare professionals make diagnoses more efficiently and accurately, and reduce medical errors and costs [48]. The development of artificial intelligence technologies has also facilitated the development of clinical decision support systems and the integration of knowledge systems in the field to solve existing problems [49]. However, while these technologies have improved the accuracy of clinical decision systems, research has also found that it is impossible to ignore the uniqueness that people value, thus driving the development of personalized medicine [50].

During the COVID-19 epidemic, artificial intelligence-based clinical decision support systems can assist medical staff in making decisions based on the patient’s actual condition, whether to hospitalize, isolate, or the patient just has a common cold [22,31,34], and even the risk of death is predicted [25]. Many countries have developed smartphone-integrated healthcare applications to collect extensive user information in order to confirm COVID-19 spread trends [23,26,32,36]. Many artificial intelligence-based clinical decision support systems, combined with graphical medicine and patients‘ electronic health records, can assist medical teams in determining screening records more accurately [30,33,39,41,42]. The electronic health records constructed by blockchain technology can be combined with smart contracts to allow medical staff to incorporate personalized patient records (such as the display of chronic diseases) in their decision-making dashboard, and to move toward personalized medicine [28].

#### 3.1.3. Telemedicine

Telemedicine refers to the provision of healthcare services and the exchange of health information at a distance [51]. During this COVID-19 epidemic, telemedicine allows patients to obtain the medical advice they need remotely, which can reduce the risk of cross-infection of mildly ill patients with seriously ill patients [52]. It can also reduce the chance of the influx of people into medical institutions, resulting in a shortage of medical resources [53]. However, the privacy of patients is a concern when using Internet-based telemedicine applications [54,55].

Telemedicine uses a smartphone or virtual reality to provide a suitable and safe environment for patients and medical staff to obtain and provide the consultation services they need [22,26,36]. The use of telemedicine can not only reduce the risk of disease transmission but effectively reduce the waste of resources, such as staff movement time [23,37,39,42]. With the contribution to the technology of wearable devices, patients can transmit their physiological data to remote medical personnel for personalized medical services, but the issue of information security is an upcoming challenge [31,32,34]. Scholars recommend encrypting technologies for transmission through blockchain technology to strengthen information security [28].

### 3.2. Human-Centered Smart Healthcare System Development

According to the literature review (Table 2), three functions (i.e., IoMT, CDSS, and Telemedicine) are the most important functions during the COVID-19 epidemic. Moreover, patient innovation should also be considered in the lifecycle of smart healthcare system development. That is, we introduced a process to develop a human-centered SHS from the sketch, with the basic functions of IoMT, CDSS, and Telemedicine. The concept of IoMT used has contributions from smart wearable devices, snoring detection, and a smart robot. As for the CDSS function, our proposed SHS can provide the patient with remote diagnoses. Through these, the cooperated healthcare unit can henceforward provide prescriptions. This is the function of telemedicine. The specific functions above will be introduced in this subsection.

This study recruited former COVID-19 patients or those with experience in caring for COVID-19 patients within a university for an academic research workshop (Figure 3). First, 46 recruited participants were divided into groups of 2–4 to discuss their previous experience of, or caring for, patients during COVID-19. These participants reflected on many of the difficulties they found in their daily lives during the treatment and isolation process after developing COVID-19. Examples include poor sleep quality at night, worries about unsubstantiated rumors on the Internet, and physical or psychological problems such as reduced exercise or social interaction due to isolation. Second, the researchers shared examples of recent research on innovative technology relating to the disease. The researchers shared the relevant applications published in the literature mentioned in the above Section 3.1. Finally, the groups were asked to report on possible future applications of these innovations in the context of their personal experience.

With this shared experience and information, the SHS prototype was proposed. When a user registers for the first time, the system will guide the user to take a simple health survey and conduct an electronic health record for the user. The user can upload personal physiological signal information obtained from the wearable device to a cloud-based database to assist medical professionals in making personalized medical suggestions. The user can also record their sleep process through smartphones. The use of artificial intelligence technology allows the user and medical personnel to determine whether there is a sleep-breathing disorder through snoring detection (Figure 4) [56].

Besides nighttime sleep healthcare, the SHS can also incorporate key-point technology to record the user’s living trajectory [57]. As shown in Figure 5a, images/videos of the care recipient in the living space can be obtained through webcams. To avoid blind spots, multiple webcams are set up to extend the image information. As shown in Figure 5b, the acquired images are input into the healthcare system (Key-point coordinates data at the bottom as shown on the right side of Figure 5b). The system detects the person and obtains the location of the key points of the human body. The system determines the location of the human body by the information on the foot position and then connects different cameras in all indoor spaces to obtain information on the movement of the human body in the lower right corner of the living space. In particular, it is necessary to reserve private spaces, such as restrooms, bedrooms, etc., to prevent users from feeling like they are being monitored when being in those spaces. During the COVID-19 epidemic, this method can be used to assist social care agencies in the trajectory tracking of confirmed case patients and prevent close contacts.

This study also includes a conversational robot to assist users in making electronic health records (Figure 6) [58]. When initially registering, users can complete their daily health records through interesting interactions with the robots. In addition, applications are built into the robots with key point detection technology [59]. Through these applications, the elderly can be assisted in physical training during home isolation. The robots can also be connected remotely, so that family members or medical teams can provide care to patients during isolation.

This study developed an SHS prototype system. People in home quarantine and healthcare practitioners can access healthcare and caring services through the system. The healthcare practitioners can obtain physiological information about their care recipients through the system, in combination with wearable devices. Voice recognition helps healthcare practitioners understand the sleep quality of their care recipients. Image recognition helps people record their activities at home. Healthcare robots allow people in home quarantine to access the social interaction and telemedicine services they need.

### 3.3. Usability Test of the SHS Prototype

The purpose of usability testing is to evaluate the efficiency and effectiveness of the system and user satisfaction through quantitative and qualitative methods [60]. The test method allows users to complete a pre-prepared operation task, and the researcher obtains users’ subjective cognitive assessments of the system through observation during the operation process and questionnaires and interviews after completing the operation task [61,62]. System developers can also use this information as a design guide for design or as a reference for improvement [63].

Thirty participants were invited to take part in the usability tests. The expert consultation process passed the National Cheng Kung University academic ethics review. The researcher explained the purpose and rights of the experiment to the participants and invited them to sign the informed consent form. The process of the usability test was ethically reviewed by the Zhongshan Medical University Hospital (CSH-2021-C-055). The Tobii eye-tracking device was used to assist in the recording of the participants during the operation of the SHS (Figure 7) [64]. At the end of the experimental session, they were interviewed by the research team. They were asked to describe the problems they had encountered during the operations by incorporating a recall-based playback method. Finally, they were invited to provide suggestions for the prototype system design [65].

Nine experts from different fields were invited to a final expert consultation. The expert consultation process passed the National Cheng Kung University Governance Framework for Human Research Ethics review (109–571). The research team conducted expert interviews using the Delphi method, and presented the system and its operation to the experts in a one-hour presentation. This allowed the experts to fully understand the design concept of the system. Then, an interview was concluded by allowing the experts to operate the system on their own, and after completing the operation they filled in the System Usability Scale (SUS) questionnaire in the form of an online questionnaire, and provided feedback on the system design. After all experts had completed the online form, the researcher then conducted a separate interview with the experts. The results of the SUS questionnaire survey received an average score of 74. The experts provided details of their recommendations. The suggestions provided by the experts were recoded according to the system framework categories and provided to future researchers and designers as a reference for design guidelines. The design guidelines for each structure of the SHS system are shown in Table 2.

The researchers invited 30 people to conduct a usability test using an eye-tracker combined with a retrospective think-aloud interview. Through usability testing with the participants, researchers can understand how to improve human–computer interactions, ease of use, and fluency. The six structures in Table 2 are the main frameworks of the prototype system. The experts provide their concerns and suggestions for these frameworks. The researcher summarized these experts’ suggestions and further discussed with them the expectations of the future system and used the KJ method to finalize the design guidelines described in Table 2 (Figure 8).

## 4. Discussion

First, we reviewed 22 SHS-related papers out of the 275 papers searched initially. The three main categories of the SHSs were defined, i.e., IoMT, CDSS, and Telemedicine. According to the literature review, we treated the three categories as the three main functions in our proposed SHS prototype. Moreover, the concept of patient innovation was also adopted to develop and evaluate the system. The design of traditional information systems often starts with the closure of the design team. This study combines patient involvement and innovation by inviting patients to participate in the system design and development process. This not only allows the system designers to be closer to the actual needs of the system users, but also shortens the time required to integrate the supply and demand sides of the system during the development process. It can also integrate the knowledge and skills that the patients may have and increase the possibilities of improving system development.

Second, as noted in the introduction and the literature review, the COVID-19 epidemic has facilitated the development of many innovative technological applications in response to the epidemic. The study provides a method to rapidly build prototype systems in order to allow system designers to review these innovative applications and confirm whether the prototype system can meet the needs of system users within a minimal timeframe. The time required for development through expert consultation and the potential risks associated with the development process are also reduced. Although the prototype system developed in this study is still rather far from actual commercial application, through an iterative development approach similar to that used in this study, these gaps can be minimized after multiple iterations, and the design team can understand whether the design deviates from the original needs of the users.

As the COVID-19 epidemic gradually slowed down, the need for such information systems as visual dashboards for COVID-19 epidemic transmission and vaccine appointment systems also gradually decreased. Researchers can use the technology and experience from the development of these systems to transfer the development of smart healthcare systems to other chronic diseases. The management of chronic diseases is another very serious topic for healthcare systems. Finally, each development process is a valuable experience for researchers and system designers. By presenting and sharing the research results, we hope to consolidate these valuable experiences and pass them on.

## 5. Conclusions

This study aims at filling the research gaps in current SHS-related studies. (1) Patient innovation: by involving the COVID-19 experienced participants, the patients’ needs can be identified; (2) Innovative technologies: studies discussing the usability of innovative technologies are few, and we focused on evaluating usability for building a more user-friendly innovative SHS; (3) Precise target users: all the participants have real experience of contracting COVID-19, interacted with the experts to provide more precise suggestions for the proposed SHS; (4) Variety of experts: experts from different professions were involved to identify issues regarding the proposed SHS. These can provide professional suggestions and recommendations to improve the system.

This paper presents a methodology for developing a prototype healthcare system that incorporates patient participation. Such an approach allows patients to be involved in the development of the prototype from the initial stages of system development and allows system developers to understand the needs of patients. Each development review process can also integrate patient and expert review to ensure that the system development process meets the needs of the end user. In addition, through actual participation in the development process, this study compiles the opinions provided by experts and their experiences to serve as design guidelines for future research and practitioners to follow. In the future, even if the COVID-19 outbreak gradually subsides, the demand from healthcare systems will not decrease. It is suggested that healthcare systems for patients with chronic diseases or non-specific diseases can be developed so that the use of smart healthcare systems can become more widespread. However, more experiments are needed to verify the validity of the design guidelines provided in this study.

## Figures and Tables

**Figure 1 healthcare-11-00847-f001:**
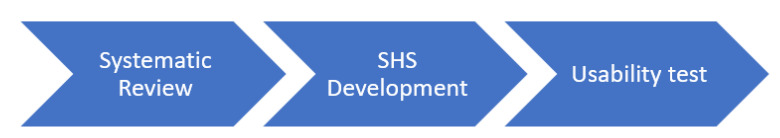
Research Flow chart.

**Figure 2 healthcare-11-00847-f002:**
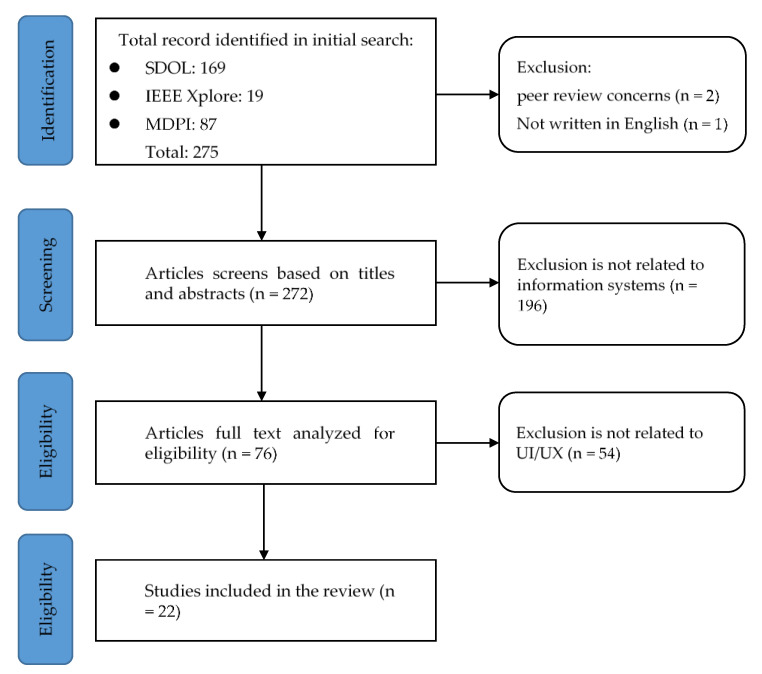
The PRISMA flow diagram.

**Figure 3 healthcare-11-00847-f003:**
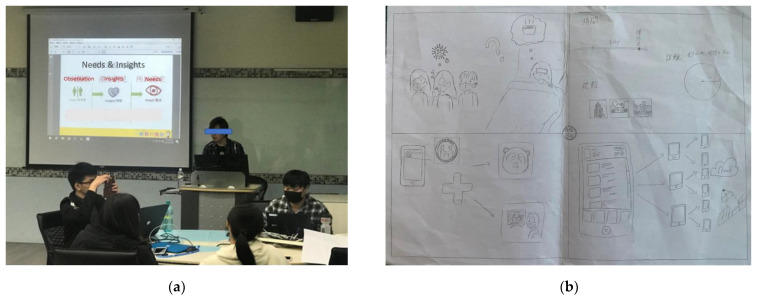
Photos of the COVID-19 workshop process. (**a**) Insights into the need for treatment or quarantine in the context of COVID-19, based on the participants’ personal experiences. (**b**) A scenario for the SHS application shared by the participants.

**Figure 4 healthcare-11-00847-f004:**
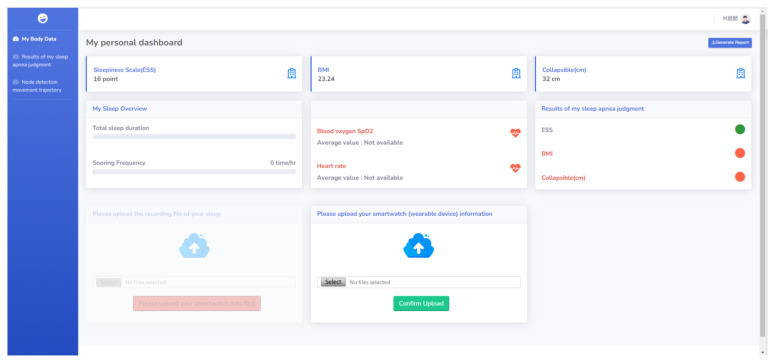
SHS sleep detection function. Users can upload the physiological information (e.g., heart rate, blood oxygen) obtained through the wearable device to the cloud database during sleep.

**Figure 5 healthcare-11-00847-f005:**
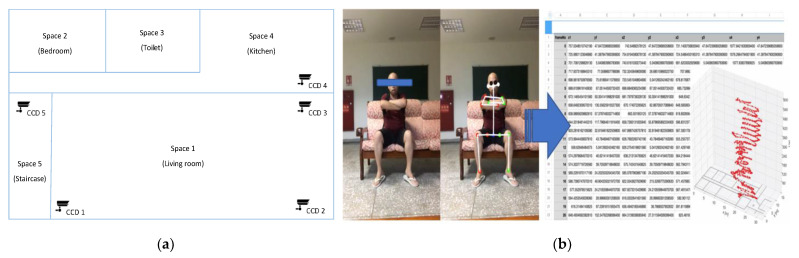
SHS life track recording function. (**a**) To avoid blind spots, multiple cameras were used for spatial information linkage. (**b**) The key point technology captures the coordinates of key points of the human body from the cameras. Then, the image information from different cameras can be linked together.

**Figure 6 healthcare-11-00847-f006:**
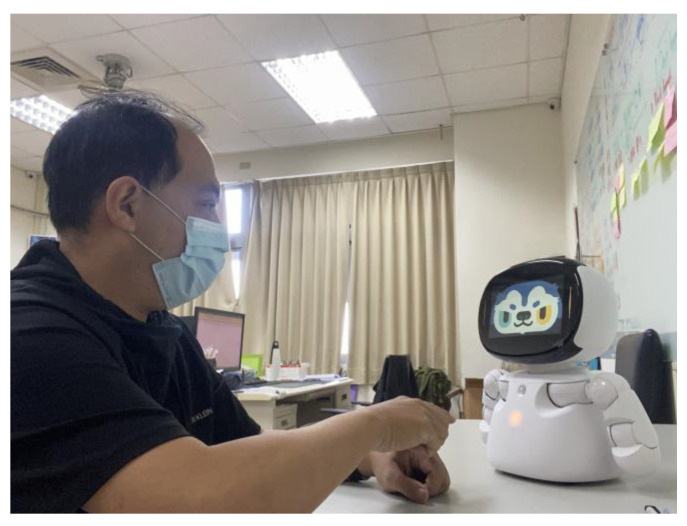
A user interacts with the talking robot. Through simple and fun interactions with the robots, users are encouraged to use these innovative technologies.

**Figure 7 healthcare-11-00847-f007:**
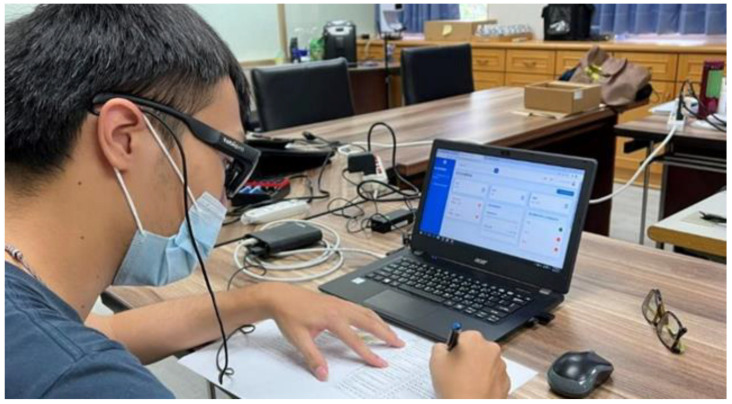
A participant wears an eye-tracking device and completes an operational task.

**Figure 8 healthcare-11-00847-f008:**
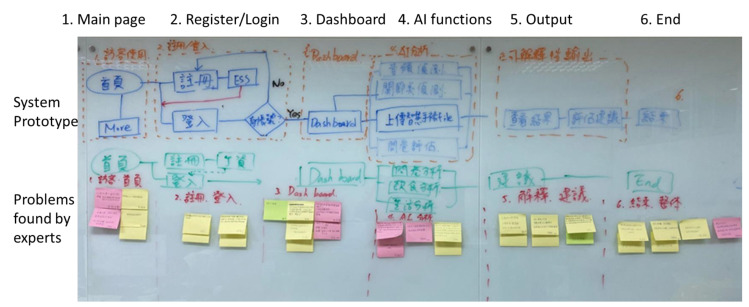
This study summarizes the design guidelines through the KJ method.

**Table 1 healthcare-11-00847-t001:** Summary table of the literature compiled for the systematic review.

No.	App Name	IoMT ^1^	CDSS ^2^	Telemedicine
Pilares, I.C.A., et al. (2022) [21]	EHRChain	√		
Abdel-Basset, M., et al. (2021) [22]	-	√	√	√
Gerli, P., et al. (2021) [23]	-	√	√	√
Rashid, M.M., et al. (2022) [24]	Block-HPCT	√		
Monjur, O., et al. (2021) [25]	-		√	
Berquedich, M., et al. (2020) [26]	-	√	√	√
Raihan, M., et al. (2022) [27]	-		√	
Subramanian, G. and A.S. Thampy (2021) [28]	NEM Blockchain	√	√	√
Ros, M. and L.S. Neuwirth (2020) [29]	Revinax	√		
Ahmed, I., et al. (2022) [30]	-		√	
Al Bassam, N., et al. (2021) [31]	-	√	√	√
Abd Elgawad, Y.Z., et al. (2022) [32]	EGYXOS	√	√	√
Greenspan, H., et al. (2020) [33]	Covictory; Health RI	√	√	
Pinto, M., et al. (2020) [34]	-		√	√
Chen, M., et al. (2022) [35]	RAMPVIS	√		
Pankhurst, T., et al. (2021) [36]	-	√	√	√
Shaikh, A., et al. (2022) [37]	Tele-COVID			√
Chang, I.-C., et al. (2022) [38]	ICU-SSS	√		
Franchini, M., et al. (2020) [39]	Dress-COV	√	√	√
Miller, E., et al. (2021) [40]	-	√		
Faezipour, M. and M. Faezipour (2020) [41]	-		√	
Naceri, A., et al. (2022) [42]	-		√	√

^1^ IoMT: The Internet of Medical Things; ^2^ CDSS: Clinical decision support system; The "√" sign indicates that the articles have used IoMT, CDSS, and telemedicine technologies and have a description of the user interface or user experience associated with these technologies.

**Table 2 healthcare-11-00847-t002:** The design guidelines adopted in this study.

App Framework	Design Guidelines	Description
Home	Graphical Imagery	The home page should not provide overly complex information but should express the concepts designed by the application in simple images.
Login/Registration	Simplification	Users are required to carefully evaluate the information they fill in, delete non-essential information fields, and simplify the registration process as much as possible.
	Automatization	Provide users with third-party login to bring in the information required to register or log in, simplifying the registration or log-in operation process
Dashboard	Timely Guidance	Users are easily confused by the complexity of the information. System designers can provide users with the assistance they need through dynamic information, such as dynamic pointer guides.
Artificial Intelligence	Explainability	It is not easy for users to understand the functions brought by artificial intelligence. The system should be designed with fast pre-processing results so that users can easily understand the effects of artificial intelligence processing.
Suggestions	Expandability	Users are easily confused by the terminology. The suggestions provided by the system should use oral terms as much as possible, and professional terms are suggested to be hyperlinked so that users can easily access the explanation of these professional terms. The system should also provide expandable links so that users can access more professional information and services they need.
Results/Feedback	Warnings	System designers should be cautious in assessing the potential risks associated with system operation. Provide users with the necessary warnings and advice when appropriate.

## Data Availability

The data presented in this study are available upon request from the corresponding author.

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
