# Peer review of "Development and Testing of the Smart Healthcare Prototype System through COVID-19 Patient Innovation"

_healthcare, 2023, doi:10.3390/healthcare11060847_

Round 1

Reviewer 1 Report

1.      “Due to the needs of design information systems to monitor the spread of epidemics, a great of research have been generated around the world. However, there has been little user-centric based research on these information systems to improve the usability of future healthcare information systems. … ” Unclear statements. The motivations become questionable. Please be more specific.

2.      Is Figure 1 from online website? Do you have the authorization to use this figure?

3.      The way of citing references in the main text should be standardized.

4.      You can mention the organization of this paper at the end of Introduction.

5.      Cite references about “the grounded theory”.

6.      You can add a flowchart to show the process of the proposed methodology.

7.      There should be a conclusions section.

Reviewer 2 Report

My comments:
1. The topic of this paper is interesting and innovative and it will contribute in related research field.

2. A section of “Related Works” or “Literature Review” is necessary for this paper.

3. The section of “Results” is very well-written.

4. A section of Conclusion” is also needed for this paper. For example, the contributions to academic research as well as theoretical implications and research limitations.

Reviewer 3 Report

After reviewing the article Development and testing of the Smart Healthcare Prototype 2 system through Covid-19 Patient Innovation, it can be seen that the objective is concrete and clear. In addition, the hypothesis proposed is intended to be achieved through research in line with the stated objectives. In this sense, it is observed after a thorough review of the article how the Smart Health System (SHS) design guidelines were compiled through a process of experts in a way that has served as a reference. All this is made concrete through the systematic review correctly proposed by the authors through which they compiled 184.

Reviewer 4 Report

This article deals with developing healthcare systems based on user feedback and provides user-centric research to develop a prototype for smart healthcare. Although the idea is novel, there are many ambiguities in the articles, and there is no uniformity in the contents. Need clarification and updates based on the following concern:

1.       What keywords were used exactly during the search process? Are there any specific criteria that only these databases were searched?

2.       How the authors came up with these subcategories, as discussed in Table 1.

3.       The major objective of the article, as discussed by the authors, is artificial intelligence-based innovative technology applications in Smart Healthcare Systems. Still, when it comes to Table 1 deals with IoMT and Telemedicine.

4.       In section 3.2 mentions that 46 participants were recruited, and researchers shared examples of recent research on innovative technology about the disease. What were those recent research innovations?

5.       What unknown factors have the researchers in Table 2 identified? These points are very basic to any human-computer interaction and applying medical information systems.

6.       There was no discussion about the perception of users about the recent technological development or what facilities or expectations of patients with any new possible system.

7.       What is the actual contribution of this article? Apart from collecting so many participants, there is no discussion about the actual feedback of participants. How can selected participants' feedback be a guiding principle for future developments?

8.       Authors have not come up with any novel contribution, and the contribution of this article is questionable. Even the discussion section does not highlight the contribution. What the authors claim as the prototype of application development is the core of any software requirements analysis to involve the system’s end users.

9.       Authors should first review the literature for user-centred medical application development and then come up with any shortcomings or missing factor that is supposed to be the core innovation of this research article. There is plenty of research available on this.

10.   Update the abstract and conclusion/discussion based on actual contribution.

11.   Clarify explicitly the feedback and keywords used in the study instead of generalised discussion and illogical pick and choose selected conditions without any justification.

Round 2

Reviewer 4 Report

The authors have sufficiently improved the article, and it can be published in the current form.

Author Response

Journal Title: Healthcare

Manuscript Title: Development and Testing of the Smart Healthcare Prototype System through Covid-19 Patient Innovation

Author(s): Po-Chih Chiu, Kuo-Wei Su*, Chao-Hung Wang, Cong-Wen Ruan, Zong-Peng Shiao, Chien-Han Tsao, Hsin-Hsin Huang

Date of the Revision: February 16, 2023.

-We sincerely appreciate all the detailed comments and suggestions made by the reviewers.

-All comments are considered and revised carefully as follows, and the modification process has been identified using Word's Track Changes feature.

Responses to Editor:

Thank you for your submission and the revised version.

Could you please reply the following comments and update the manuscript, if necessary?

Comments:

1) The methodology has 3 phases. I, systematic review; II, human-centered smart healthcare system development; III, usability test.

  1. a) What is the aim of SHL system? Why is it related to COVID?

Response:

Descriptions of Smart Healthcare System is added to subsection 1, as follows. (Page 3, Introduction, Line 113, 120-123)

“COVID-19 virus is highly infectious and has high variability.”

“This study aims to combine smartphones and webcams with artificial intelligence to enable healthcare practitioners to provide remote care to people in home quarantine. The researchers hope that the study can reduce the risk of contact for healthcare practitioners and improve their working quality and efficiency.”

  1. b) What exactly is being tested in this SHL system? (sleep detection, spatial information, talking robot, eye traking? What is the purpose?

Response:

Descriptions of Smart Healthcare System is added to subsection 3.2, as follows. (Page 11, Human-Centered Smart Healthcare System Development, Line 371-378).

“This study developed an SHS prototype system. People in home quarantine and healthcare practitioners can access healthcare and caring services through the system. The healthcare practitioners can obtain physiological information about their care recipients through the system in combination with wearable devices. Voice recognition helps healthcare practitioners understand the sleep quality of their care recipients. Image recognition helps people record their activities at home. Healthcare robots allow people in home quarantine to access the social interaction and telemedicine services they need.”

  1. c) What was the outcome of the usability test?

Response:

Descriptions of the usability test is added to subsection 3.3, as follows. (Page 13, Human-Centered Smart Healthcare System Development, Line 420-424).

“The researchers invited 30 people to conduct a usability test using an eye tracker and a retrospective think-a-loud interview. Through usability testing with the participants, researchers can understand how to improve human-computer interactions, ease of use, and fluency.”
